# Cu Nanoparticle-Loaded Nanovesicles with Antibiofilm Properties. Part I: Synthesis of New Hybrid Nanostructures

**DOI:** 10.3390/nano10081542

**Published:** 2020-08-06

**Authors:** Lucia Sarcina, Pablo García-Manrique, Gemma Gutiérrez, Nicoletta Ditaranto, Nicola Cioffi, Maria Matos, Maria del Carmen Blanco-López

**Affiliations:** 1Department of Chemistry, Università degli Studi di Bari Aldo Moro, via Orabona 4, 70125 Bari, Italy; lucia.sarcina@uniba.it (L.S.); nicoletta.ditaranto@uniba.it (N.D.); nicola.cioffi@uniba.it (N.C.); 2Department of Physical and Analytical Chemistry, University of Oviedo, Julián Clavería 8, 33006 Oviedo, Spain; garciampablo@uniovi.es (P.G.-M.); gutierrezgemma@uniovi.es (G.G.); 3Department of Chemical and Environmental Engineering, University of Oviedo, Julián Clavería 8, 33006 Oviedo, Spain; 4Instituto Universitario de Biotecnología de Asturias, University of Oviedo, 33006 Oviedo, Spain

**Keywords:** hybrid nanostructures, nanovesicles, nanoparticles, copper, antibiofilm

## Abstract

Copper nanoparticles (CuNPs) stabilized by quaternary ammonium salts are well known as antimicrobial agents. The aim of this work was to study the feasibility of the inclusion of CuNPs in nanovesicular systems. Liposomes are nanovesicles (NVs) made with phospholipids and are traditionally used as delivery vehicles because phospholipids favor cellular uptake. Their capacity for hydrophilic/hydrophobic balance and carrier capacity could be advantageous to prepare novel hybrid nanostructures based on metal NPs (Me-NPs). In this work, NVs were loaded with CuNPs, which have been reported to have a biofilm inhibition effect. These hybrid materials could improve the effect of conventional antibacterial agents. CuNPs were electro-synthesized by the sacrificial anode electrolysis technique in organic media and characterized in terms of morphology through transmission electron microscopy (TEM). The NVs were prepared by the thin film hydration method in aqueous media, using phosphatidylcholine (PC) and cholesterol as a membrane stabilizer. The nanohybrid systems were purified to remove non-encapsulated NPs. The size distribution, morphology and stability of the NV systems were studied. Different quaternary ammonium salts in vesicular systems made of PC were tested as stabilizing surfactants for the synthesis and inclusion of CuNPs. The entrapment of charged metal NPs was demonstrated. NPs attached preferably to the membrane, probably due to the attraction of their hydrophobic shell to the phospholipid bilayers. The high affinity between benzyl-dimethyl-hexadecyl-ammonium chloride (BDHAC) and PC allowed us to obtain stable hybrid NVs c.a. 700 nm in diameter. The stability of liposomes increased with NP loading, suggesting a charge-stabilization effect in a novel antibiofilm nanohybrid material.

## 1. Introduction

A biofilm is commonly defined as a complex community of microorganisms adherent on a surface and organized in a polymeric matrix, usually containing exopolysaccharide [1,2,3]. These structures have a heterogenic network of aqueous compartments interposed between glycocalyx-enclosed microorganism in stalk- or mushroom-like structures [3], holding more than one single microbial species. The biofilm resistance issue could be considered phenotypic. Bacteria in biofilm present a higher antimicrobial resistance (AMR) compared to planktonic forms [3].

Prevention and destruction of biofilms is a challenging task requiring knowledge from several scientific research branches. Due to AMR mechanisms, there is an urgent need to search for novel efficient formulations. A reported effective strategy to develop more efficient materials to prevent/minimize biofilm formation is surface modification through the addition of metal particles (Cu, Zn, Ti, Ag) to polymeric blends [4,5,6,7,8] or biopolymers such as starch solutions [9]. Recent studies reported that carbon/copper nanoparticles (CuNP) hybrids were effective as biocides, with the carbon component in charge of capturing the bacteria and the CuNPs responsible for bacteria destruction. Moreover, the outer carbon layers protected the metallic copper from external oxidation [10]. It has been proved in the literature that, in order to have bactericidal inhibitory effect, a surface should be covered by relatively hydrophobic polymers that are positively charged [3].

Metal nanoparticles (Me-NPs) such as AuNPs, AgNPs or CuNPs display unique properties that make them suitable at surface science for the development of antimicrobial formulations. Previous studies investigated the integration of NPs in biomaterials showing unique recognition, catalytic, and inhibition properties [6,11,12,13,14]. Especially gold or silver NPs are largely used for biomedical application since they can be used as biomarkers and drug-delivery agents with a bactericidal effect [15]. The antibacterial action towards different microorganisms exerted by polymer nanocomposites loaded with copper NPs has already been proved [4,16,17,18,19,20,21,22], along with the possibility to tune the ionic copper release. It is possible to achieve efficient disinfection by simply changing the CuNP loading and/or the thickness and formulation of the surfactant shell. Indeed, as evident from the plots reported in Figure 1, when a stabilizer with four butyl alkyl chains is used to prepare CuNPs, the released plateau copper amount is around 1ppm. This value decreases as the length of the surfactant alkyl chains is increased, along with the slowdown of the kinetics and the kinetic constants. This could be explained by the increase in the shell thickness, which is able to modulate the copper ions’ release from the surface of the NP core [21,23,24,25]. Moreover, in quaternary ammonium compounds (QAC), both the fatty-acyl chain length distributions and the degree of C–C saturation will significantly affect antimicrobial activity, with a maximized action against Gram positive bacteria when chain length is n = 12–14 and against Gram negative bacteria when n = 14–16 [26].

Some of the reported modes of action of anti-biofilm molecules are inhibition via interference in the quorum sensing pathways, the adhesion mechanism, disruption of extracellular DNA, protein, lipopolysaccharides, exopolysaccharides and the secondary messengers involved in various signaling pathways [27]. It has been proved that novel nanostructures involving copper fusion increased antimicrobial activity against biofilms. This seems to be due to the release of copper ions, which inhibited the quorum sensing in *Methylobacterium spp*. This resulted in inhibition of the expression of the genes that form biofilms [14].

Nanovesicles (NVs) are self-assembled structures of lipids enclosed in bilayers forming single (unilamellar) or concentric membranes (multilamellar), which divide hydrophilic and hydrophobic compartments. They can be considered as “soft nanoparticles” because of the interaction of their components resembling biological systems [28]. The most explored NV systems are liposomes and niosomes, which differ in terms of the components used for the NV membrane layer formation. The components mostly used are cationic lipids such as DOTMA (trimethyl [2,3-(dioleyloxy) propyl] ammonium chloride) or DPPC (DL-dipalmitoylphosphatidylcholine) and the nonionic sorbitan monostearate or trioleate, which resemble the structure of phospholipids that form biological membranes [28,29]. The choice of nanovesicle type affects the final shape and thickness of the membrane as well as supramolecular reorganization, permeability, elasticity and compatibility with biological materials [30,31,32,33,34,35].

Several techniques have been implemented to prepare NVs, depending on the specific requirement and technologic purposes. The procedure typically involves the following steps: drying down lipids from organic solvent, re-dispersing them in aqueous media, purifying liposomes, analyzing the morphology of the product and measuring, when possible, the encapsulation efficiency [36,37,38]. The selected method could affect some morphological factors such as size and size distribution, the physical instability of vesicles and their encapsulation efficiency.

Liposomes are meant to be stable and bio-related structures for the transport and the controlled release of drugs. Thus, the investigation of hybrid NP-loaded NVs could combine the bactericidal effect of metal NPs enforcing the efficacy of NVs for permeability and delivery. Park et al. (2005) demonstrated the formulation of gold-loaded liposomes including 3–4 nm NPs into lipid (DPPC) bilayers. Similar results were obtained for silver NP inclusion [15,39]. Moreover, it was found that the encapsulation of AgNPs into liposomes enhanced antibacterial efficacy, reducing the concentration of NPs necessary to achieve a complete inhibition of bacterial growth [40]. On the other hand, it has just recently been demonstrated that liposomes functionalized with quaternary ammonium compounds are able to inhibit bacterial adherence and biofilm formation [33]. Therefore, it could be expected that the combination of NVs and CuNPs in hybrid nanostructures could enhance antibiofilm properties.

Over the recent decades, the capability of using electrochemical methods for the synthesis of transition metal NPs [23,41], and its peculiarity to have notable control on particle size thanks to voltage control has been proved. The so-called sacrificial anode electrolysis (SAE) is based on the oxidation of the bulk anode-material and the subsequent reduction of the formed metal ions in the presence of a stabilizer in order to obtain core-shell NPs dispersed in the electrosynthesis solution.

The aim of the part I of this work was to study how NPs can be loaded in liposomes and to examine the behavior of these novel composite structures in terms of bilayers stability and fluidity. Their real antimicrobial capacity will be tested in part II. CuNPs were electro-synthesized by the SAE technique in organic media and characterized in terms of morphology through transmission electron microscopy (TEM). To assess NP inclusion in the lipid bilayer, the NVs were prepared by the thin film method in aqueous media, using phosphatidylcholine (PC) and cholesterol as a membrane stabilizer. This method promotes the formation of multilamellar vesicle structure [28,37], and it was chosen for this study to synthesize robust NVs with mechanical strength. The hybrid NP-loaded NVs were purified and their morphology was characterized by TEM. The particle size distribution and colloidal stability of the suspension were studied by dynamic light scattering (DLS), ζ-potential and multiple lights scattering (MLS). This study will open the path to novel nanohybrid materials with improved antibiofilm inhibition properties.

## 2. Materials and Methods

### 2.1. Materials

For CuNP preparation: two different surfactants were tested: tetra-dodecyl ammonium chloride (TDoAC), with four symmetric dodecyl alkyl chains, and benzyl-dimethyl-hexadecyl ammonium chloride (BDHAC), with four asymmetric alkyl chains. Both alkylammonium salts, chloroform (CHCl_3_) and tetrahydrofuran (THF), were purchased from Merck KGaA (Darmstadt, Germany).

For vesicle preparation, the materials used were as follows: phosphatidylcholine (PC), from soybean (Phospholipon 90G) and cholesterol. PC was a kind gift from Lipoid (Ludwigshafen am Rhein, Germany), and cholesterol was purchased from Merck KGaA (Darmstadt, Germany).

### 2.2. Preparation of CuNPs

The synthesis of copper colloids was performed in a three-electrode cell equipped with a copper anode, a platinum cathode and a reference electrode made of Ag/AgNO_3_ (0.1 M in Acetonitrile). The electrodes were dipped in a solution containing the surfactant which acted both as an electrolyte and a capping agent for NPs, providing a stable shell that prevented particle agglomeration, and avoided cathode metallization. Chloroform and tetrahydrofuran were used as solvents to let CuNPs disperse in the NVs. Two different surfactants were tested: tetra-dodecyl ammonium chloride (TDoAC), and benzyl-dimethyl-hexadecyl ammonium chloride (BDHAC).

The electrosynthesis was performed using the potentiostat CHI-1140 B (CH Instruments Inc., Austin, TX, USA). During the synthesis, an amperometric I-t curve was recorded, from which it was possible to read the total charge value useful to calculate the process yield. Finally, the colloid was characterized in order to determine particle size and morphology. At least three replicates were carried out for each experiment.

### 2.3. Characterization of CuNPs

Morphological analysis of the copper colloids was performed using an Tecnai Spirit G2 electron microscope (FEI Company, Hillsboro, OR, USA) with a LaB6 filament as its electron source (120 kV operating voltage). The transmission electron microscopy (TEM) images were used to measure the size and size distribution of the CuNPs using ImageJ software (Wayne Rasband (NIH), Bethesda, MD, USA). The histograms were prepared with SigmaPlot 12.0, plotting the frequency counts of the NP diameters vs. the NP size.

### 2.4. Preparation of Empty NVs and NP-Loaded NVs

The preparation of both empty NVs and hybrid CuNP-loaded NVs was carried out by using the thin film hydration method [37,42]. This well-established method for niosome or liposome technology is suitable for the effective encapsulation of additives (such as drugs) in the hydrophobic compartment [43]. Natural soybean phosphatidylcholine (PC) was used as the main lipid component and cholesterol as the membrane stabilizer. Several molar ratios were tested as reported in Table 1.

A Rotary evaporator Buchi R-205 was used for the preparation of both the empty and hybrid NVs. Firstly, the empty NVs were prepared by a 5 mM solution of PC in a mixture of chloroform and THF (CHCl_3_:THF) in a 9:1 (*v*/*v*) ratio. Organic solvents were evaporated when setting the rotary evaporator. Experiments by modifying the temperature, rotation speed, evaporation time, and vacuum pressure were tested. After the evaporation of solvents, the film was left at 150 mbar under rotation for 1 h to enhance thinner formation, then the flask was filled with nitrogen and stored at room temperature overnight. The transparent and homogeneous film formed was hydrated by adding deionized water. The temperature of the deionized water, rotation speed and hydration time were optimized accordingly.

Table 1 shows the experimental parameters tested for the inclusion of both CuNPs stabilized by TDoAC (Cu@TDoAC) and by BDHAC (Cu@BDHAC).

### 2.5. Purification of NP-Loaded NVs

After the hydration, a purification step was implemented to separate CuNP–NV hybrids from non-encapsulated NPs. A red lipophilic dye (Fat Red Bluish) 0.5% (*w*/*w*) was used to better visualize the vesicles in a gel filtration PD10 empty column, in-house packed with Sepharose CL-4B. The Sepharose was kept at room temperature, then a solution was prepared with 13 mL of gel in 15 mL of milli-Q water. The solution was poured and left to pack over a 0.45 µm polyethylene frit previously washed in ethanol, and was then inserted in the column. Additionally, 2.5 mL of the sample was poured in the column; milli-Q-filtered water was added subsequently. The first 3 mL of the sample that dropped out, corresponding to the purified vesicles, was collected.

### 2.6. Characterization of Empty NVs and NP-Loaded NVs

Vesicle and hybrid colloid morphologic characterization was assessed with a JEM-1011 transmission electron microscope (TEM, JEOL, Akishima–Tokyo, Japan). The samples were diluted in a 1:100 ratio before placing a drop of the diluted suspension on a copper grid (TAAB, carbon coated 300 mesh). Since the NVs were transparent to the electron beam, it was necessary to perform a negative staining of the samples. After placing the samples containing the NVs on the grid, a drop of 2% (*w*/*w*) phosphotungstic acid solution was added on the grid and the excess was removed after ten seconds. The contact time with the staining agent was kept short enough to avoid an excessive darkening of the samples.

A Malvern Zetasizer Nano-ZS instrument, (Malvern, UK) was used for determining both size distribution of the empty NVs and hybrids by dynamic light scattering (DLS), and to estimate ζ-potential by laser Doppler velocimetry (LDV), which provides information concerning the stability of the empty liposomes and hybrid CuNP-loaded NVs. The samples were diluted in a 1:100 volume and the test was replicated three times for each composition, monitoring the stability over one week at 25 °C.

A Formulaction Turbiscan lab expert with an ageing station was also used to test the colloidal stability by verifying creaming, flocculation and precipitation in ageing conditions. Empty NVs prepared with PC and cholesterol were tested, as well as the hybrid NVs loaded with CuNPs and stabilized by BDHAC, with three different concentrations of the loaded copper colloids. The analysis lasted six days with a sampling time of two hours, and the samples were stored in the ageing station at 38 °C.

## 3. Results

### 3.1. Synthesis and Morphological Characterization of CuNPs

During the electrosynthesis of copper colloids, the current vs. time curve was recorded, hence the charge value. Comparing these experimental data with the weight change of electrodes, it was possible to estimate the percentage yield of the process and the concentration of copper in the final dispersion. Cu@TDoAC NPs were prepared using the surfactant concentration and the overpotential value already adjusted for this process (0.1 M, 1.5 V) [21].

When the asymmetric salt BDHAC was used as a stabilizer, the electrochemical parameters did not lead to the same optimal results, and fast precipitation and NP aggregation were observed. Therefore, an increased surfactant concentration was used with the aim of inducing a greater stabilizing effect and balancing the smaller steric hindrance of BDHAC as a shell-forming component [44]. Moreover, THF was tested as a single solvent to prevent phase separation during CuNP inclusion in the NVs.

Figure 2 reports the TEM images and the size distribution histograms of the Cu@TDoAC and Cu@BDHAC colloids.

In both cases, the images showed the presence of almost spherical NPs and an average diameter of a few nanometers with a homogeneous in-plane distribution, proving that the selected electrochemical parameters effectively yielded CuNPs with the desired morphology.

### 3.2. Preparation and Characterization of Empty NVs

The morphology of the empty NVs, studied through TEM analysis, is shown in Figure 3. Multilamellar structures (zoomed insert) and spherical structures with an average size of 100–1000 nm obtained as reported in literate for similar working conditions were observed [45,46]. The DLS characterization of the empty NVs presented only one peak, with a mean Z-average size of 1.6 ± 0.1 µm with a polydispersity index (PDI) of 0.18 ± 0.03. Some variations in size were observed after ageing; indeed, the NVs displayed an aggregation trend and precipitated after few hours as two peaks distribution appeared, with mean sizes of 1.90 ± 0.02 µm and 5.48 ± 0.01 µm. Therefore, PDI increased to 0.31 ± 0.01. NV ζ-potential measured by LDV was always found to be negative for all the replicates at c.a. −30 mV. As reported in the literature, a strong charged layer (positive or negative) can increase electrostatic repulsion between particles, thus, a value generally lower than −30 mV or higher than +30 mV for ζ-potential guarantees suspension stability [47].

### 3.3. Preparation and Characterization of NP-Loaded NVs

Cu@TDoAC showed the best colloidal stability and narrower size distribution, hence, it was chosen for the first NP inclusion test. On the other hand, Cu@BDHAC was selected—despite some partial aggregation occurring after a few days—because of its widely proven biocompatibility and antimicrobial properties [21], to exploit the synergistic effect with copper [44] when inserted in the NVs. The Cu@TDoAC- and Cu@BDHAC-loaded NVs were characterized and compared in order to further investigate a possible antibiofilm action exerted by the CuNPs combined with the disinfection properties of the quaternary ammonium compounds (part II). The preparation conditions for hybrid vesicles with Cu@TDoAC loading were optimized in test 3 (refer to Table 1). The characterization of this sample is reported in Figure 4 in regard to both TEM images and DLS analysis. The result was a dispersion of NVs with a wide size distribution in the 100–1000 nm diameter range, confirmed by both the characterization techniques. Some agglomeration also occurred, as an evident shift in the peaks’ position was recorded during the three replicate DLS analyses. An adhesion of NPs limited to the surface of the NVs could be assumed, and/or some affinity between TDoAC and PC could lead to the agglomeration of hybrid systems [48].

LDV analysis gave a ζ-potential value of 29 ± 2 mV for the Cu@TDoAC–NVs prepared with the highest Cu concentration (test 3), suggesting a positive charging of the sample due to the presence of copper NPs. Moreover, the ageing of the hybrids obtained in these conditions showed a broadening on the size distribution over time and a decrease in ζ-potential, whose value was measured after one week at 21 ± 4 mV.

The inclusion of the Cu@BDHAC colloid in lipid bilayers, tested in three different conditions (as stated in Table 1), was optimized by using colloids synthetized with THF as the sole solvent, which guarantees a uniform dispersion and controlled evaporation. Furthermore, the pressure control allowed a moderate evaporation speed, leading to a well-dried and transparent final film. This insured a more efficient hydration step and an NV dispersion in the sub-micron range with a narrow size distribution.

Figure 5 shows the TEM characterization of these hybrids, for which the formation of complex structures could be observed. NPs appeared to be assembled in clusters, only partially included in the NVs (Figure 5a,b). The presence of copper clusters in the range of 50–100 nm was confirmed by TEM analysis performed without negative staining (Figure 5c). This evidence could be explained by NP aggregation due to low stabilization of BDHAC and/or to a possible interaction of the surfactant with PC due to their similar molecular structure. Some works in the literature also suggest aggregation as a way for hybrids to minimize the strain in the bilayer caused by NP inclusion [49,50]. Moreover, from the DLS characterization (Figure 5d), two main peaks were observed centered at 91 nm and 712 nm. As the size of the empty NVs has been estimated to be in the micron range, it seems reasonable that the main DLS peak of the NP-loaded vesicles (712 nm) was probably attributable to the hybrids, while the peak at 91 nm could be ascribed to the free CuNP clusters.

The filtration in the Sepharose-4B gel column was made to better understand if this peak assignment was confirmed and if an exclusion of cluster was possible. The red-full line in the graph (Figure 5d) refers to the DLS analysis after filtration. The purification excluded the smaller structures leading only higher sized vesicles to be selected. ζ-potential was evaluated for hybrids holding Cu@BDHAC colloids; a comparison between the empty and hybrid NVs is presented in Table 2. A more positive shift upon the increasing of the Cu colloid concentration loaded in the precursor solution was observed, as expected when introducing a positively charged species.

Finally, colloidal stability of the empty NVs and hybrid suspension (Cu@BDHAC–NVs) was tested through the backscattering profile method acquired for non-filtered samples.

In Figure 6, the backscattering profiles are shown for the empty NVs and hybrid NVs with the highest concentration of the loaded Cu (test 3, corresponds to c.a. 72 µM for the used copper colloid).

The empty NVs (Figure 6a) presented a high peak at the bottom of the cell, indicating precipitation behavior after few hours of ageing, whereas hybrid nanocolloids did not exhibit precipitation. By comparing the backscattering profiles at different copper concentrations loaded into the hybrids (Figure 7), their suspension stability was confirmed. Indeed, small variation in backscattering values was observed along the cell, indicating a slight increase in the particle size with time. Nevertheless, the backscattering percentage after 6 days remains stable through the cell height, especially for a LIP:NPs ratio of 800:1. This means that an improved stability was obtained when increasing the copper loading.

The increasing colloidal stability observed for the highest CuNP loading could be related to major steric hindrance, as well as a sufficient charge-stabilizing effect induced by positively charged nanoparticles. Another major difference between the empty and hybrid NV is membranes fluidity. As the presence of NPs could increase fluidity, to avoid their subsequent escape, the presence of a membrane stabilizer is needed. The choice of cholesterol was essential, since it intercalates between lipid chains, balancing the increased volume of polar heads of membrane components [50,51]. The resulting hybrid assembly show a promising stability, keeping its structure over ageing conditions without releasing NP clusters in the solution.

## 4. Conclusions

In the present work, the synthesis of hybrid systems based on electro-synthetized CuNPs was investigated. The largely known antibacterial properties of CuNPs and their application in materials and life science provided us the opportunity to work on new hybrid nanosystems with the aim of exploiting biochemical and physical properties of materials such as liposomes.

The inclusion of CuNPs stabilized by different quaternary ammonium compounds in vesicular systems made of phosphatidylcholine (PC) was demonstrated, indicating that the hydrophobic shell of the NP has a good affinity for vesicle bilayers.

Benzyl-dimethyl-hexadecyl-ammonium chloride (BDHAC) as a stabilizing surfactant of CuNPs was tested with the aim of developing a potentially synergistic antibacterial agent, combining in the same vesicle the effects of copper ions and BDHAC. A high affinity between this ammonium salt and PC was noticed, obtaining stable hybrid NVs 700–1000 nm in diameter. The stability of liposomes increased with NP loading with a maximum Cu concentration of 72 µM, suggesting a combined hindrance and charge-stabilization effect confirmed by dynamic light scattering, ζ-potential measurements and backscattering-monitored precipitation.

Preliminary experiments on antibiofilm efficacy were performed by testing our hybrid NPs against the proliferation of *Staphylococcus aureus* (Gram +), *Pseudomonas aeruginosa* (Gram −), and *Candida parapsilosis* (fungus). The inhibition effect was poor for the CuNP loading that we tested, therefore, further investigation will be carried out to assess the proper hybrid NV composition.

The application of the Cu-loaded nanosystems herein proposed is promising and will be investigated in bio-medical routes. The whole release of the NP cluster enclosed in the NVs could be avoided. Furthermore, these hybrid nanovesicles could provide a way to control the ionic release of copper through liposomes, which are already used for drug transport, and counteract the strong resistance of biofilms, subsequently leading to their destruction.

The application of our hybrid synergistic systems in the fight against antibiotic-resistant biofilms is envisaged, mostly in the cases where the use of vesicular systems might be beneficial when compared with the use of unprotected nanophases. While safety regulations are generally posing severe limits to the use of ultrafine NPs [52], the same does not apply to NVs with sizes in the range of 100 to 1000 nm, like those of the present study.

## Figures and Tables

**Figure 1 nanomaterials-10-01542-f001:**
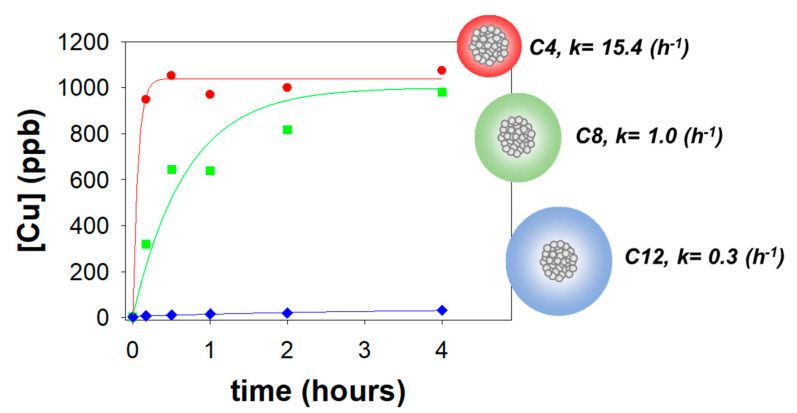
Effect of the alkyl chain length of the stabilizer on the copper release concentration and on the kinetic constant for copper nanocomposites with the same copper nanoparticle (CuNP) %*w*/*w* loading. Red: tetra butyl ammonium chloride (C4); green: tetra octyl ammonium chloride (C8); blue: tetra dodecyl ammonium chloride (C12).

**Figure 2 nanomaterials-10-01542-f002:**
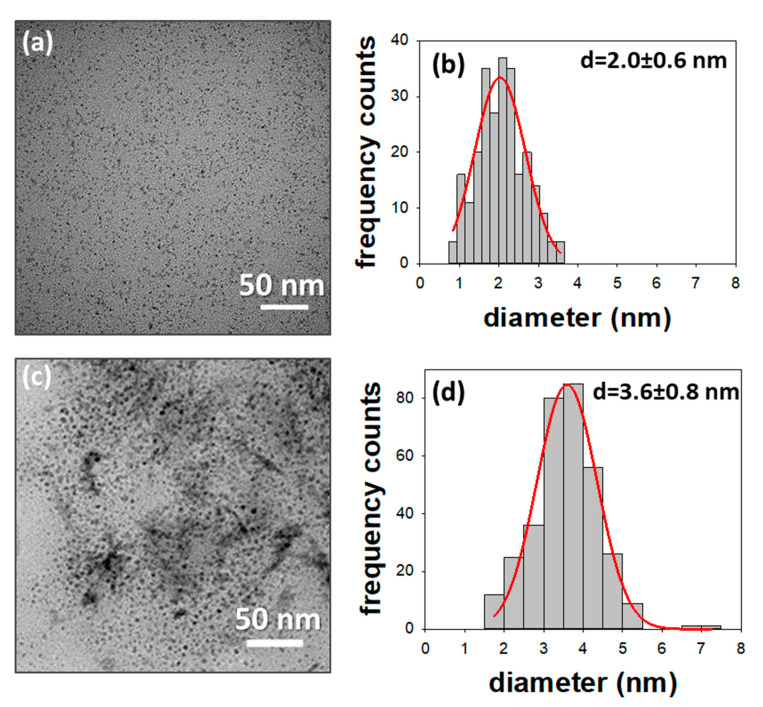
Transmission electron microscopy (TEM) images of the Cu@TDoAC colloid (**a**), and the Cu@BDHAC colloid (**c**), along with the size distribution histograms (**b**,**d**). Sample (**a**) d = 2.0 ± 0.6 nm, n = 250; sample (**c**) d = 3.6 ± 0.8 nm, n = 330.

**Figure 3 nanomaterials-10-01542-f003:**
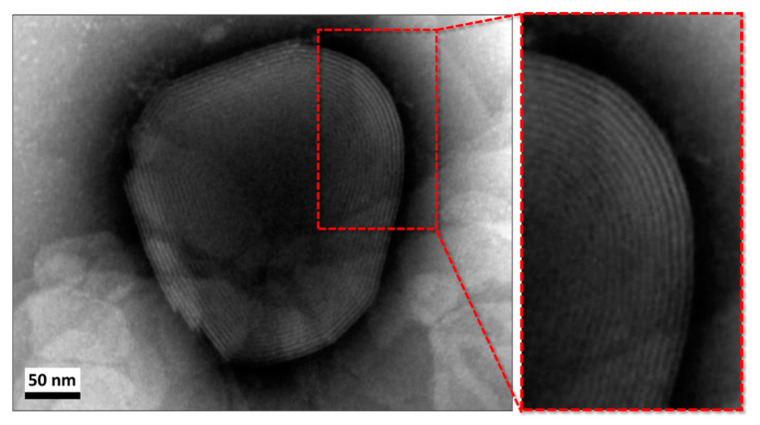
TEM images of the empty NVs. The inset highlights the obtained multilamellar structure.

**Figure 4 nanomaterials-10-01542-f004:**
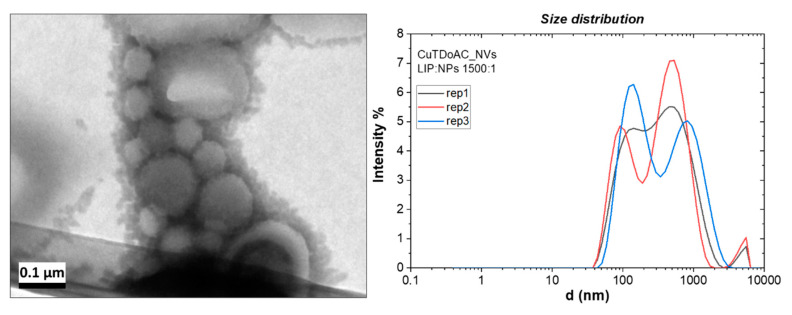
TEM image (**left**) and three dynamic light scattering (DLS) (**right**) replicate measurements of the Cu@TDoAC–NV hybrids prepared at optimized conditions: relative amount of lipid to nanoparticles(LIP:NPs) ratio 1500:1 *w*/*w*.

**Figure 5 nanomaterials-10-01542-f005:**
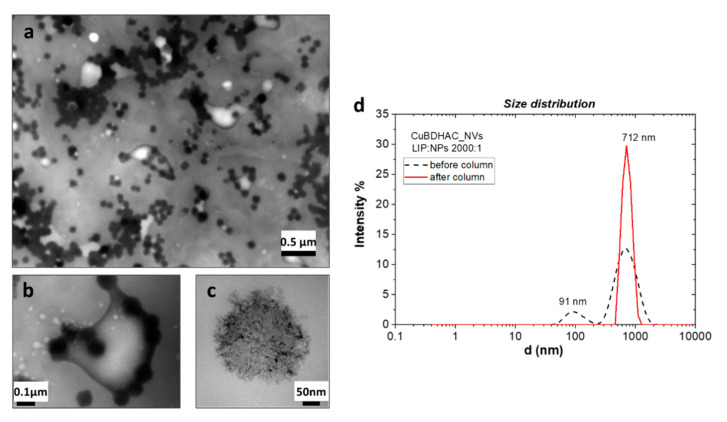
TEM image (**left**) and DLS (**right**) of the Cu@BDHAC–NV hybrids prepared at optimized conditions (LIP:NPs ratio 2000:1 *w*/*w*). Overview of complex structures (**a**) and zoom on a hybrid vesicle (**b**) and on one NP cluster (**c**). Size distribution recorded before and after sample filtration (**d**).

**Figure 6 nanomaterials-10-01542-f006:**
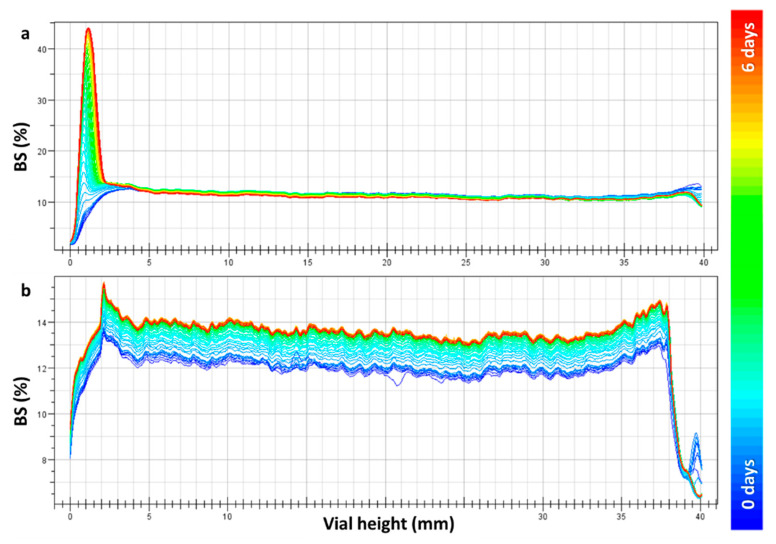
Backscattering profile vs. vials height acquired over 6 days. Signal comparison of the empty NVs (**a**) and Cu@BDHAC–NP-loaded NV hybrids (**b**) at a LIP:NPs ratio of 800:1 *w*/*w*.

**Figure 7 nanomaterials-10-01542-f007:**
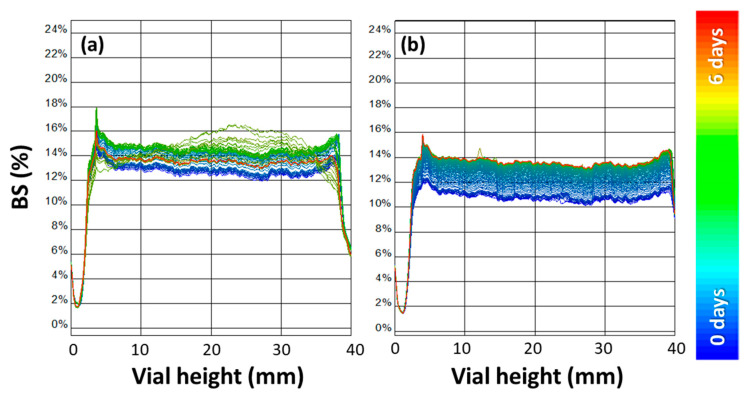
Backscattering profile (%) vs. vials height (mm) acquired over 6 days. Signal of the Cu@BDHAC–NP-loaded NV hybrids at a LIP:NPs ratio of 2000:1 *w*/*w* (**a**) and at a LIP:NPs ratio of 1200:1 *w*/*w* (**b**).

**Table 1 nanomaterials-10-01542-t001:** Experimental parameter setting for the inclusion in nanovesicles (NVs) of CuNPs stabilized by tetra-dodecyl ammonium chloride (TDoAC—Cu@TDoAC) and benzyl-dimethyl-hexadecyl-ammonium chloride (BDHAC—Cu@BDHAC). The ratio of membrane components, phosphatidylcholine and cholesterol (PC:CHO), their lipid concentration and relative amount of lipid to nanoparticle composition (LIP:CuNPs) are listed for each test. Specifications of the solvents used, and temperature (T), pressure (P), rotation speed (R), and time (t) for evaporation and hydration steps are also reported.

**TDoAC (Cu@TDoAC)**												
	**Membrane Components**			**Solvents**		**Evaporation**				**Hydration P_ATM_**		
	**PC:CHO Molar Ratio**	**Lipid (mM)**	**LIP:CuNPs (*w*/*w*)**	**CHCl_3_:THF (*v*/*v*)**	**Vol (mL)**	**T °C**	**P (mbar)**	**R (rpm)**	**t (min)**	**T °C**	**R (rpm)**	**t (min)**
Test1	1:0	10	500:1	9:1	20	41	140	140	40	60	150	20
Test2	2:1	10	700:1	9:1	30	50	490	150	60	60	150	20
Test3	2:1	10	1500:1	9:1	15	52	180	150	90	55	150	30
**BDHAC (Cu@BDHAC)**												
	**Membrane Components**			**Solvents**		**Evaporation**				**Hydration P_ATM_**		
	**PC:CHO Molar Ratio**	**Lipid (mM)**	**LIP:CuNPs (*w*/*w*)**	**CHCl_3_:THF (*v*/*v*)**	**Vol (mL)**	**T °C**	**P (mbar)**	**R (rpm)**	**t (min)**	**T °C**	**R (rpm)**	**t (min)**
Test1	2:1	5	2000:1	9:1	25	53	150	150	60	50	150	5
Test2	2:1	5	1200:1	9:1	25	53	150	150	90	50	150	5
Test3	2:1	5	800:1	9:1	25	53	150	150	30	50	150	5

**Table 2 nanomaterials-10-01542-t002:** Comparison of ζ-potential results obtained by laser Doppler velocimetry for the empty NVs and non-filtered hybrids with three increasing concentration of Cu@BDHAC.

SAMPLE	LIP:NPs (*w*/*w*)	ζ-Potential (mV)
**NVs**	--	−24 ± 5
**Hybrid 1**	2000:1	−11 ± 4
**Hybrid 2**	1200:1	5 ± 4
**Hybrid 3**	800:1	21 ± 6

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
