# Peer review of "Cu Nanoparticle-Loaded Nanovesicles with Antibiofilm Properties. Part I: Synthesis of New Hybrid Nanostructures"

_nanomaterials, 2020, doi:10.3390/nano10081542_

Round 1

Reviewer 1 Report

Dearest Authors,

i regret to say that your work, although potentially interesting, is currently far from being considered for publication. serious major flaws are present, IMHO:

paper main focus lacks a clear definition of potential application, including also a description and discussion of its potential realistic applicability considereing regulatory and normative framework

experimental design misses all in vitro testing to assess antibacterial efficacy, at least preliminary

discussion is poor

bibliography is by far not sufficiently up-to-date and this also reflects in the poor introduction and discussion

best

Author Response

Response to Reviewer 1 Comments

Dearest Authors,

I regret to say that your work, although potentially interesting, is currently far from being considered for publication. serious major flaws are present, IMHO:

Point 1. Paper main focus lacks a clear definition of potential application, including also a description and discussion of its potential realistic applicability considering regulatory and normative framework

Response 1. A sentence has been added in the Conclusions section discussing perspective applications.

Point 2. Experimental design misses all in vitro testing to assess antibacterial efficacy, at least preliminary

Response 2. Preliminary results on the antibacterial efficacy of hybrid systems have been mentioned, while further studies are under investigation. The in vitro testing needs an extensive analysis that could not be included in the present work

Point 3&4. Discussion is poor. Bibliography is by far not sufficiently up-to-date and this also reflects in the poor introduction and discussion

Response 3&4. We thank the referee for his/her valuable comments. In the revised version we have added new and updated references to the introduction. We have also rewritten some parts to better specify the aim of the work and its impact. Some sentences have been added to improve the discussion of the results.

List of added references:

  1. Chari, N.; Felix, L.O.; Davoodbasha, M.A.; Sulaiman Ali, A.; Nooruddin, T. In vitro and in vivo antibiofilm effect of copper nanoparticles against aquaculture pathogens. Biocatal. Agric. Biotechnol. 2017, 10, 336–341, doi:10.1016/j.bcab.2017.04.013.
  2. Galli, R.; Hall, M.C.; Breitenbach, E.R.; Colpani, G.L.; Zanetti, M.; de Mello, J.M.M.; Silva, L.L.; Fiori, M.A. Antibacterial polyethylene - Ethylene vinyl acetate polymeric blend by incorporation of zinc oxide nanoparticles. Polym. Test. 2020, 89, 106554, doi:10.1016/j.polymertesting.2020.106554.
  3. Eltz, F.Z.; Vebber, M.C.; Aguzzoli, C.; Machado, G.; Da Silva Crespo, J.; Giovanela, M. Preparation, characterization and application of polymeric thin films containing silver and copper nanoparticles with bactericidal activity. J. Environ. Chem. Eng. 2020, 8, 103745, doi:10.1016/j.jece.2020.103745.
  4. Khan, Z.; Al-Thabaiti, S.A. Biogenic silver nanoparticles: Green synthesis, encapsulation, thermal stability and antimicrobial activities. J. Mol. Liq. 2019, 289, 111102, doi:10.1016/j.molliq.2019.111102.
  5. Chen, H.; Wu, J.; Wu, M.; Jia, H. Preparation and antibacterial activities of copper nanoparticles encapsulated by carbon. New Carbon Mater. 2019, 34, 382–389, doi:10.1016/s1872-5805(19)30023-x.
  6. Beeton, M.L.; Aldrich-Wright, J.R.; Bolhuis, A. The antimicrobial and antibiofilm activities of copper(II) complexes. J. Inorg. Biochem. 2014, 140, 167–172, doi:https://doi.org/10.1016/j.jinorgbio.2014.07.012.
  7. Seo, Y.; Hwang, J.; Lee, E.; Kim, Y.J.; Lee, K.; Park, C.; Choi, Y.; Jeon, H.; Choi, J. Engineering copper nanoparticles synthesized on the surface of carbon nanotubes for anti-microbial and anti-biofilm applications. Nanoscale 2018, 10, 15529–15544, doi:10.1039/c8nr02768d.
  8. Roy, R.; Tiwari, M.; Donelli, G.; Tiwari, V. Strategies for combating bacterial biofilms: A focus on anti-biofilm agents and their mechanisms of action. Virulence 2018, 9, 522–554.
  9. Montefusco-Pereira, C. V; Formicola, B.; Goes, A.; Re, F.; Marrano, C.A.; Mantegazza, F.; Carvalho-Wodarz, C.; Fuhrmann, G.; Caneva, E.; Nicotra, F.; et al. Coupling quaternary ammonium surfactants to the surface of liposomes improves both antibacterial efficacy and host cell biocompatibility. Eur. J. Pharm. Biopharm. 2020, 149, 12–20, doi:https://doi.org/10.1016/j.ejpb.2020.01.013.
  10. Cui, H.; Li, W.; Li, C.; Vittayapadung, S.; Lin, L. Liposome containing cinnamon oil with antibacterial activity against methicillin-resistant Staphylococcus aureus biofilm. Biofouling 2016, 32, 215–225, doi:10.1080/08927014.2015.1134516.
  11. Ardizzone, A.; Blasi, D.; Vona, D.; Rosspeintner, A.; Punzi, A.; Altamura, E.; Grimaldi, N.; Sala, S.; Vauthey, E.; Farinola, G.M.; et al. Highly Stable and Red-Emitting Nanovesicles Incorporating Lipophilic Diketopyrrolopyrroles for Cell Imaging. Chem. – A Eur. J. 2018, 24, 11386–11392, doi:10.1002/chem.201801444
  12. Marchianò, V.; Matos, M.; Serrano-Pertierra, E.; Gutiérrez, G.; Blanco-López, M.C. Vesicles as antibiotic carrier: State of art. Int. J. Pharm. 2020, 585.
  13. Yang, S.-T.; Kreutzberger, A.J.B.; Lee, J.; Kiessling, V.; Tamm, L.K. The role of cholesterol in membrane fusion. Chem. Phys. Lipids 2016, 199, 136–143, doi:https://doi.org/10.1016/j.chemphyslip.2016.05.003.

Reviewer 2 Report

Dear Editor,

dear Authors,

I have gone through the manuscript and I like this work. The investigated systems are interesting but the paper needs minor revision before publication.

1- Introduction needs to correlate a bit more with objective of the manuscript.

2- Some new recent and relevant reference need to be incorporated

3- Please check sentence in Abstract section:

"The nano hybrid systems were purified to remove unbounded NPs and characterized in terms of size and size distribution, morphology and stability through zeta potential and stability."

4- Table 1 is not easy to read. (titles are not ok in my version)

5- line 177: Do you mean the Laser doppler velocimetry (LDV) coupled in the malvern zetasizer? It seems to me confusing. In my opinion a zetasizer is a coupled device between a DLS  and a LDV. Please revise I the MS that you are using the Malvern zetasizer for both measurements…. Z average and zeta potential. If not, please explain.

6- Size distribution. Do you have just information in intensity % (Figure 4 and figure 5)? Do you have some data in volume % or in number%? Would be interesting for the discussion.

At the present state I suggest a minor revision.

Author Response

Response to Reviewer 2 Comments

Dear Authors,

I have gone through the manuscript and I like this work. The investigated systems are interesting but the paper needs minor revision before publication.

Point 1. Introduction needs to correlate a bit more with objective of the manuscript.

Response 1. The introduction has been modified in the revised version of the manuscript to better specify the aim of the work. The objective has been more clearly defined.

Point 2. Some new recent and relevant reference need to be incorporated

Response 2. New recent and relevant references have been included. 

List of added references:

  1. Chari, N.; Felix, L.O.; Davoodbasha, M.A.; Sulaiman Ali, A.; Nooruddin, T. In vitro and in vivo antibiofilm effect of copper nanoparticles against aquaculture pathogens. Biocatal. Agric. Biotechnol. 2017, 10, 336–341, doi:10.1016/j.bcab.2017.04.013.
  2. Galli, R.; Hall, M.C.; Breitenbach, E.R.; Colpani, G.L.; Zanetti, M.; de Mello, J.M.M.; Silva, L.L.; Fiori, M.A. Antibacterial polyethylene - Ethylene vinyl acetate polymeric blend by incorporation of zinc oxide nanoparticles. Polym. Test. 2020, 89, 106554, doi:10.1016/j.polymertesting.2020.106554.
  3. Eltz, F.Z.; Vebber, M.C.; Aguzzoli, C.; Machado, G.; Da Silva Crespo, J.; Giovanela, M. Preparation, characterization and application of polymeric thin films containing silver and copper nanoparticles with bactericidal activity. J. Environ. Chem. Eng. 2020, 8, 103745, doi:10.1016/j.jece.2020.103745.
  4. Khan, Z.; Al-Thabaiti, S.A. Biogenic silver nanoparticles: Green synthesis, encapsulation, thermal stability and antimicrobial activities. J. Mol. Liq. 2019, 289, 111102, doi:10.1016/j.molliq.2019.111102.
  5. Chen, H.; Wu, J.; Wu, M.; Jia, H. Preparation and antibacterial activities of copper nanoparticles encapsulated by carbon. New Carbon Mater. 2019, 34, 382–389, doi:10.1016/s1872-5805(19)30023-x.
  6. Beeton, M.L.; Aldrich-Wright, J.R.; Bolhuis, A. The antimicrobial and antibiofilm activities of copper(II) complexes. J. Inorg. Biochem. 2014, 140, 167–172, doi:https://doi.org/10.1016/j.jinorgbio.2014.07.012.
  7. Seo, Y.; Hwang, J.; Lee, E.; Kim, Y.J.; Lee, K.; Park, C.; Choi, Y.; Jeon, H.; Choi, J. Engineering copper nanoparticles synthesized on the surface of carbon nanotubes for anti-microbial and anti-biofilm applications. Nanoscale 2018, 10, 15529–15544, doi:10.1039/c8nr02768d.
  8. Roy, R.; Tiwari, M.; Donelli, G.; Tiwari, V. Strategies for combating bacterial biofilms: A focus on anti-biofilm agents and their mechanisms of action. Virulence 2018, 9, 522–554.
  9. Montefusco-Pereira, C. V; Formicola, B.; Goes, A.; Re, F.; Marrano, C.A.; Mantegazza, F.; Carvalho-Wodarz, C.; Fuhrmann, G.; Caneva, E.; Nicotra, F.; et al. Coupling quaternary ammonium surfactants to the surface of liposomes improves both antibacterial efficacy and host cell biocompatibility. Eur. J. Pharm. Biopharm. 2020, 149, 12–20, doi:https://doi.org/10.1016/j.ejpb.2020.01.013.
  10. Cui, H.; Li, W.; Li, C.; Vittayapadung, S.; Lin, L. Liposome containing cinnamon oil with antibacterial activity against methicillin-resistant Staphylococcus aureus biofilm. Biofouling 2016, 32, 215–225, doi:10.1080/08927014.2015.1134516.
  11. Ardizzone, A.; Blasi, D.; Vona, D.; Rosspeintner, A.; Punzi, A.; Altamura, E.; Grimaldi, N.; Sala, S.; Vauthey, E.; Farinola, G.M.; et al. Highly Stable and Red-Emitting Nanovesicles Incorporating Lipophilic Diketopyrrolopyrroles for Cell Imaging. Chem. – A Eur. J. 2018, 24, 11386–11392, doi:10.1002/chem.201801444
  12. Marchianò, V.; Matos, M.; Serrano-Pertierra, E.; Gutiérrez, G.; Blanco-López, M.C. Vesicles as antibiotic carrier: State of art. Int. J. Pharm. 2020, 585.
  13. Yang, S.-T.; Kreutzberger, A.J.B.; Lee, J.; Kiessling, V.; Tamm, L.K. The role of cholesterol in membrane fusion. Chem. Phys. Lipids 2016, 199, 136–143, doi:https://doi.org/10.1016/j.chemphyslip.2016.05.003.

Point 3. Please check sentence in Abstract section:

"The nano hybrid systems were purified to remove unbounded NPs and characterized in terms of size and size distribution, morphology and stability through zeta potential and stability."

Response 3. The sentence has been rewritten:

“The nano hybrid systems were purified to remove non-encapsulated NPs. The size distribution, morphology and stability of the NVs systems were studied.”

Point 4. Table 1 is not easy to read. (titles are not ok in my version)

Response 4. The text format of Table 1 has been adjusted, as well as its description, to be easily readable.

Point 5. line 177: Do you mean the Laser doppler velocimetry (LDV) coupled in the malvern zetasizer? It seems to me confusing. In my opinion a zetasizer is a coupled device between a DLS and a LDV. Please revise I the MS that you are using the Malvern zetasizer for both measurements…. Z average and zeta potential. If not, please explain.

Response 5. The sentence concerning LDV has been modified according to the Reviewer suggestion, to clarify that DLS and LDV were both performed through the same Zetasizer instrument.

See line 227-230 in the manuscript: “A Malvern Zetasizer Nano-ZS instrument, Malvern, UK was used for determining both size distribution of empty NVs and hybrids by Dynamic Light Scattering (DLS), and to estimate the ζ-potential by Laser doppler velocimetry (LDV), which gives information concerning the stability of empty liposomes and hybrid CuNPs loaded NVs.”

Point 6. Size distribution. Do you have just information in intensity % (Figure 4 and figure 5)? Do you have some data in volume % or in number%? Would be interesting for the discussion.

Response 6. The results given by the Zetasizer software are presented only in intensity percentage of detected particles for each correspondent dimensions (i.e. diameter size in nm on the x-axis). Although the volume or number of NVs could give more information, for the present study we can discuss size distribution only in terms of relative intensities.

Reviewer 3 Report

The subject of this research is very interesting as it presents a method of Synthesis of New Nano-Hybrid Structures Based on 2 Cu Nanoparticle-Loaded Nanovesicles. Drug resistance is becoming one of the most common and urgent problems of world public health and this type of research can provide useful tools for the scientific community.

I advise the authors to make revisions to improve the clarity of the presentation of the results obtained without making new experiments.

Regarding the introduction, I suggest that the authors treat in more detail the properties of the encapsulated Cu in different ways, for example by carbon. I kindly ask the authors to expand the part concerning from line 54 to 58 not limiting themselves to a reference list but treating the topic in more detail.

Please insert in the materials and methods section the description of how the histograms of the NPs size distributions were made.

I think the work could be much clearer if the results were integrated, also presenting the figures of the supplementaries in the main test

I would also advise colleagues to reformat table 1, possibly reducing the font to make the column headings more readable

I also recommend modifying figure 2 by putting the same ranges on the axes ... for y, from 0 to 100, for x the interval from 0 to 8 with a step of 0.5

The caption of figure 5 must be rewritten referring to all the panels. In the same way in the text it is necessary to comment on all the panels. Also the comments on panel b of figure 6 are lost in the text.

Author Response

Response to Reviewer 3 Comments

The subject of this research is very interesting as it presents a method of Synthesis of New Nano-Hybrid Structures Based on 2 Cu Nanoparticle-Loaded Nanovesicles. Drug resistance is becoming one of the most common and urgent problems of world public health and this type of research can provide useful tools for the scientific community.

I advise the authors to make revisions to improve the clarity of the presentation of the results obtained without making new experiments.

Point 1. Regarding the introduction, I suggest that the authors treat in more detail the properties of the encapsulated Cu in different ways, for example by carbon. I kindly ask the authors to expand the part concerning from line 54 to 58 not limiting themselves to a reference list but treating the topic in more detail.

Response 1. In the revised version of the manuscript more detailed examples of several types of systems used to encapsulate metallic particles with antibiofilm activity have been included.

Point 2. Please insert in the materials and methods section the description of how the histograms of the NPs size distributions were made.

Response 2. A sentence has been added in the Experimental section explaining how the histograms of the NPs size distributions were made.

Point 3. I think the work could be much clearer if the results were integrated, also presenting the figures of the supplementary in the main test.

Response 3. We thank the Reviewer for the suggestion. The figure presented in the supplementary information has been included in the main text, renumbered as Figure 7. The description and the discussion of this figure compared to Figure 6 has been implemented accordingly.

Point 4. I would also advise colleagues to reformat table 1, possibly reducing the font to make the column headings more readable

Response 4. The text format of Table 1 has been revised to be clearer and more readable.

Point 5. I also recommend modifying figure 2 by putting the same ranges on the axes ... for y, from 0 to 100, for x the interval from 0 to 8 with a step of 0.5

Response 5. The histograms have been obtained plotting the frequency counts of the NPs diameters vs. the NPs size. For this reason, it is not convenient to modify the range for y axis because it strongly depends on the frequency distribution of the NPs diameters, disregarding the total number of particles. The x axis ranges have been changed to be both from 0 to 8 nm.

Point 6. The caption of figure 5 must be rewritten referring to all the panels. In the same way in the text it is necessary to comment on all the panels. Also the comments on panel b of figure 6 are lost in the text.

Response 6. The caption of Figure 5 has been modified, by checking that all the panels are explained. The discussion of Figure 6 have been adjusted in the main text.

Round 2

Reviewer 1 Report

Dearest Authors,

i aprreciate the heavy rewritten paper. Given experimental data on bacterial-contrast efficacy are just preliminary menioned, I would suggest you update the work underlining this aspect and eventually changing the title accordingly (for example adding a "part I: ...").

best

Author Response

Response 1. Thank you for the reconsideration of the article. The authors agree that mentioning a second part of the work, where the antimicrobial capacity of the synthetized hybrid materials is investigated, could clarify the aim of the present paper. Accordingly, the title has been changed, and few changes have been made in the introduction and discussion section.

All the text variations are highlighted by track of changes in the revised version of the manuscript.